# Programmed multimaterial assembly by synergized 3D printing and freeform laser induction

Bujingda Zheng[1], Yunchao Xie[1], Shichen Xu[2], Andrew C. Meng [3], Shaoyun Wang[1], Yuchao Wu[1], Shuhong Yang[4], Caixia Wan[4], Guoliang Huang[1], James M. Tour [2,5,6] & Jian Lin [1] ✉

In nature, structural and functional materials often form programmed three-dimensional (3D) assembly to perform daily functions, inspiring researchers to engineer multifunctional 3D structures. Despite much progress, a general method to fabricate and assemble a broad range of materials into functional 3D objects remains limited. Herein, to bridge the gap, we demonstrate a freeform multimaterial assembly process (FMAP) by integrating 3D printing (fused filament fabrication (FFF), direct ink writing (DIW)) with freeform laser induction (FLI). 3D printing performs the 3D structural material assembly, while FLI fabricates the functional materials in predesigned 3D space by synergistic, programmed control. This paper showcases the versatility of FMAP in spatially fabricating various types of functional materials (metals, semiconductors) within 3D structures for applications in crossbar circuits for LED display, a strain sensor for multifunctional springs and haptic manipulators, a UV sensor, a 3D electromagnet as a magnetic encoder, capacitive sensors for human machine interface, and an integrated microfluidic reactor with a built-in Joule heater for nanomaterial synthesis. This success underscores the potential of FMAP to redefine 3D printing and FLI for programmed multimaterial assembly.

Assembly of multimaterials into structural and functional components is ubiquitous in nature, inspiring researchers to explore new design principles and fabrication methodologies for creating engineered three-dimensional (3D) structures with multifunctionalities[1,2]. Traditionally, hybridized fabrication techniques can be used to achieve the goal[3–6], but they require multiple, subsequent processes. For instance, producing multilayer 3D printed circuit boards (PCBs) entails steps of etching, lamination, heated pressing, drilling, etc.[7]. The processes require high capital investment while generating unwanted waste streams, thus posing a significant challenge to sustainability. To enhance material utilization efficiency and circumvent the challenge of assembling multimaterials, several new technologies such as mechanics-driven assembly, transfer printing, and multimaterial 3D printing have been emerged recently[8,9].

Among these emerging techniques, the multimaterial 3D printing has attracted attention for its potential benefits, including cost-effectiveness, reduced waste generation, and easy customization. Thus, many strides have been recently made for applications in multimaterial fabrication[9]. For example, a direct ink writing (DIW) method enables to fabricate 3D soft electronics[6] and light emitting diodes

[1]Department of Mechanical and Aerospace Engineering, University of Missouri, Columbia, MO 65201, USA. [2]Department of Chemistry, Rice University, Houston 77005 TX, USA. [3]Department of Physics and Astronomy, University of Missouri, Columbia, MO 65201, USA. [4]Department of Chemical and Biomedical Engineering, University of Missouri, Columbia, MO 65201, USA. [5]Department of Materials Science and Nano Engineering, Rice University, 6100 Main Street, Houston 77005 TX, USA. [6]Smalley-Curl Institute, Rice University, 6100 Main Street, Houston 77005 TX, USA. ✉e-mail: linjian@missouri.edu

(LEDs)[10]. Embedded 3D printing has facilitated production of flexible sensors by embedding functional carbon grease within a polymer encapsulation[11]. A multi-nozzle DIW printer with a rapid material switching capability has been developed to print diverse wax-based structures[12]. Further advancement in a core-shell DIW nozzle has enabled assembled multimaterials, such as epoxy/silicone, into different 3D structures, including a sandwiches[13] and helices[14]. Multi-axis fused filament fabrication (FFF)[15] and conformal DIW[16] can make conformal deposition of conductive filaments onto 3D curved surfaces. Moreover, there have been other noteworthy developments in the field like digital light processing (DLP) of multimaterials[17–19].

However, within the realm of multimaterial fabrication, these techniques still face challenges of lacking versatility in precisely placing functional materials within 3D structures and access to broader material options. For instance, the embedded 3D printing necessitates preparation of a mold for the structural materials[11]. This necessity imposes constraints on the capability of achieving complex geometries, such as in hollow and freestanding features. In the case of core-shell 3D printing, although it can print objects with inner structures made from functional materials, the functional and structural materials are extruded simultaneously and continuously, so depositing the functional materials in predesigned locations, such as outer surface, is not achievable[13,14]. Besides the limitation in the complexity of printed structures, they often suffer from limited materials options. For instance, the multi-nozzle DIW extrude composite inks that contains both electrically conductive materials and polymers[12,15], rendering the resulting materials with low electrical conductivity and low mechanical strength. DLP is quite limited to photosensitive resins[17]. Moreover, the process for multimaterial printing requires switching between different vats while purging non-polymerized residual materials out from the vats, which results in inefficient materials utilization. All these challenges underscore the need for further innovation in the multimaterial fabrication methodologies with improved versatility in the structure complexity and broadened materials choices.

Direct laser writing (DLW) has shown versatility in patterning various functional materials through induced photothermal or/and photochemical effects[20]. They significantly expand the library of available materials ranging from laser-induced graphene (LIG)[21,22], to metals[23], metal oxides[24], semiconductors[25], and ceramics[26]. A recent trend in DLW is to assemble these functioning materials into 3D structures[27,28], while this goal is largely limited by its capability in fabricating the functional materials on 2D planes. Recently, we introduced a freeform laser induction (FLI) method facilitated by a 5-axis laser processing platform. This method enables the direct fabrication of 3D conformable electronics on freeform surfaces[29]. While this technique represents an advancement in DLW capabilities, spatially patterning functional materials within predesigned locations of 3D structures to create multifunctional objects remains a challenge.

Here, to tackle this challenge, we present a freeform multimaterial assembly process (FMAP) that synergistically marries advantages of three techniques—FLI, DIW, and FFF—to seamlessly assemble both structural and laser-processable functional materials into 3D engineered objects with complex geometries and multifunctionalities. FFF can construct structural components from commercially available thermoplastics such as polycarbonate (PC), polyethylene terephthalate glycol (PETG) and thermoplastic polyurethane (TPU), and polyvinylidene fluoride (PVDF), while FLI selectively converts the FFF-printed material into LIG in predesigned position in the 3D space. DIW can deposit precursors onto LIG electrodes for later laser-inducing other functional materials, e.g., silver, iron, cobalt, nickel, and copper oxides, to obtain LIG-based functional composites. With the advantages of FFF and FLI, the functional materials are either encapsulated inside the printed 3D objects or on their outside surfaces, thus forming integrated functioning 3D devices. They include a crossbar circuit for a light emitting diode (LED) array, strain sensors for an integrated multifunctional spring and a haptic manipulator, a UV sensor, a 3D electromagnet as a rotational encoder, a capacitive sensor for human machine interface (HMI), and an integrated microfluidic reactor with a built-in Joule heater for nanomaterial synthesis. The demonstrated methodology shows a series of advances. Firstly, it facilitates programmed assembly of both functional and structural materials into the integrated 3D devices by a single apparatus, thus eliminating the requirement of many processing steps in different apparatuses. Secondly, it augments the versatility by direct laser processing of different functional materials with negligible precursor waste streams. Thirdly, FLI decouples the synthesis of the functional materials from FFF and DIW, thus it can pattern them in any predesigned locations of the 3D structures. Overall, this methodology represents a step forward in the creation of integrated, multifunctional 3D objects with applications across electronics/sensors, HMI, robotics, and functional microfluids.

## Results

FMAP, as depicted in Fig. 1a, has 5 degrees of freedom (DOF) by incorporating three linear motions and two rotational motions. Motors connected to harmonic gear boxes provide sufficient torque for the rotational axes with precise movements. It has two additional motors that control the extrusion of FFF and DIW. Figure 1b depicts the three end effectors: an FFF end, a DIW nozzle, and a laser module. They are configured as follows. Both the FFF and DIW nozzles are assembled with the laser module to save space. The FFF end is placed in parallel to the laser module, while the DIW nozzle is installed alongside the FFF that is strategically rotated 15° counterclockwise from the Z axis. When operating the extrusion by DIW, the A-axis motor rotates 15° clockwise, aligning the DIW syringe parallel to the z axis, while the FFF end is rotated away. This configuration prevents contact between the extruded ink and the FFF end. The role of the laser is to convert the FFF printed materials into LIG and the DIW deposited ink into functional materials, such as semiconductors, metals, and metal oxides. The laser module emits light at a wavelength of 450 nm with a maximum power of 5 W.

We use fabrication of a 3D wireless LED as an example to explain the manufacturing workflow by FMAP (Fig. 1c) with details shown in Supplementary Fig. 1 and Supplementary Movie 1. The process starts with FFF of a few layers of a PC structure. Then, the laser is turned on to selectively induce the PC to a LIG electrode. Next, an Ag precursor (silver citrate) is deposited onto the LIG electrode by DIW. Another laser induction converts the Ag precursor to Ag infiltrated in the LIG matrix to obtain a highly conductive LIG/Ag electrode, on top of which new PC layers are printed by FFF. During the laser induction, the laser is controlled in the five DOF to conformally pattern any complex geometry of the electrode onto the printed 3D structures. Figure 1d displays the fabricated wireless LED, with a cross-section view illustrating the distribution of the conductive LIG/Ag electrode inside the PC structure. When powered with a charging coil, the fabricated LED is on as intended (Supplementary Movie 2). To induce LIG from non-laser-convertible polymers such as TPU and PETG, an ink consisting of lignin and silver citrate is first deposited on the selective positions of the FFF-printed TPU structure. Since the build plate is heated at 100 °C, the solvent in the deposited ink evaporates rapidly. The laser induction on the ink can be operated immediately without stop, leading to formation of a LIG/Ag composite. This altered process is illustrated in Supplementary Fig. 2. Owing to the flexible nature of TPU and the LIG/Ag electrode, the same 3D wireless LED can be conformally fabricated onto a flexible cloth substrate. Figure 1e and Supplementary Movie 3 display that the fabricated flexible 3D LED maintains good lighting performance when wirelessly powered.

In addition to Ag, other materials are synthesized via laser induction to afford diverse functionalities of the 3D structures. For instance, Fe can be incorporated for magnetism. Energy-dispersive spectrometry (EDS) was conducted to analyze the spatial distribution

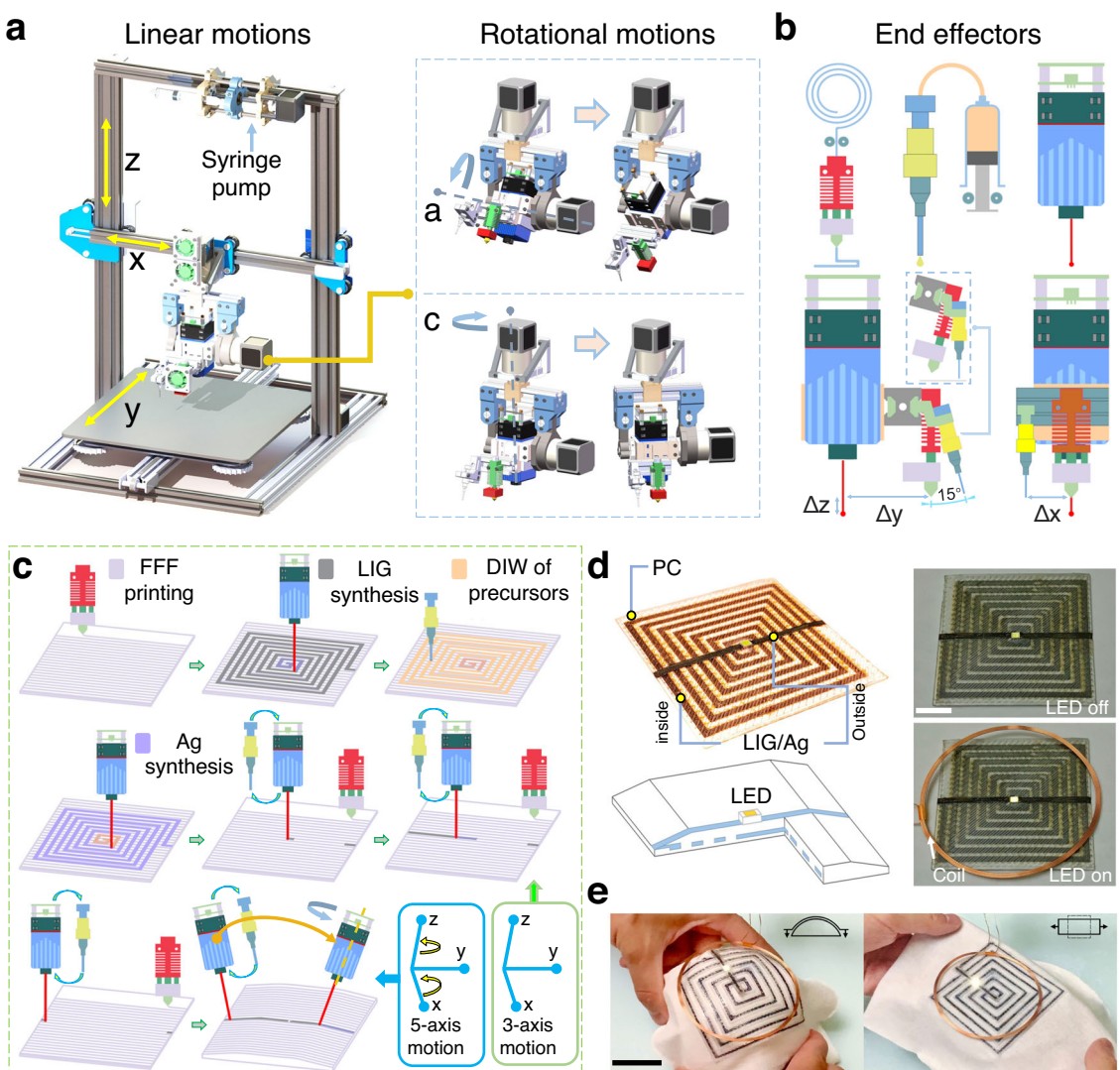

**Fig. 1 | Schematic of FMAP platform and workflow of fabricating 3D devices by assembling structural and functional materials using FMAP. a** Schematic showing FMAP platform and its linear and rotational actuation system. **b** Schematic of end effectors for FFF, DIW, and FLI, as well as the installation offset between these three end effectors. **c** Workflow of fabricating a 3D wireless LED circuit with LIG (induced from PC) and Ag electrodes by FMAP. To distinguish the resulting materials from different processes, the FFF 3D printing results are colored light purple, LIG conductive traces are colored grey, the precursor of silver is colored light orange, and the silver is colored light blue. **d** Scheme and a photograph of the fabricated 3D wireless LED. **e** A photograph of a fabricated 3D wireless LED with LIG (induced from lignin) and Ag electrodes on a cloth, being pressed onto a convex object and stretched. Scale bar: 10 mm.

of the synthesized metals (Ag, Fe, Co, and Ni) and a metal oxide (CuO) within LIG induced from different polymers (PC and PETG). Figure 2a illustrates that LIG is highly porous, agreeing well with the results reported in our previous works[21,30]. The synthesized metals and metal oxides are in a form of nanoparticles (NPs) well dispersed inside the LIG matrix as depicted in the elemental mapping of the composites. Figure 2b illustrates the cross-sectional scanning electron microscopy (SEM) images collected from four regions of the LIG embedded 3D structures. The printing layer heights vary from 0.1 to 0.3 mm, with the laser operated at 2.5 W, in a focused status, and with a scan rate of 300 mm/min. The examined regions encompass: a pure polymer, a LIG region, an area where LIG overlays a polymer, and a polymer region with LIG underneath. The images show a clear polymer gap in between the LIG layers when the layer height exceeds 0.15 mm, implying an incomplete conversion of the entire layer into LIG. This observation agrees well with the result shown in Supplementary Fig. 3, where the electrical resistance in the z-axis direction is dramatically increased when the layer height exceeds 0.15 mm. Supplementary Fig. 4 shows that a slower scan rate results in a smaller sheet resistance, reaching

the smallest value of 98.2 Ω/sq at 100 mm/min. The relationship between the LIG thickness and laser power is revealed in Supplementary Fig. 5. It shows that as the laser power rises, the LIG thickness increases. The laser induction resolution is illustrated in Fig. 2c, where a conductive LIG trace with a width of 200 μm can effectively power an LED. Supplementary Fig. 6 indicates that the linewidth of the laser induced functional materials varies based on the precursors and laser parameters with the best one achieving ~100 μm in the silver. Tensile testing specimens (dimensions: 25 mm × 3 mm × 1 mm) with embedded LIG in the center (dimensions: 25 mm × 2 mm × 0.4 mm) were produced by our FMAP. The PC was printed with the layer heights of 0.1–0.2 mm.

Figure 3a-i shows that their tensile strengths all exceed 35 MPa, which is compatible to pure PC specimens, indicating well-maintained mechanical properties even if the PC is partially converted to LIG. Furthermore, tensile testing was performed on specimens embedded with LIG. The LIG dimensions were varied while keeping laser power and printing layer height constant. Supplementary Fig. 7 shows that as the LIG thickness and width increases, respectively, both the tensile

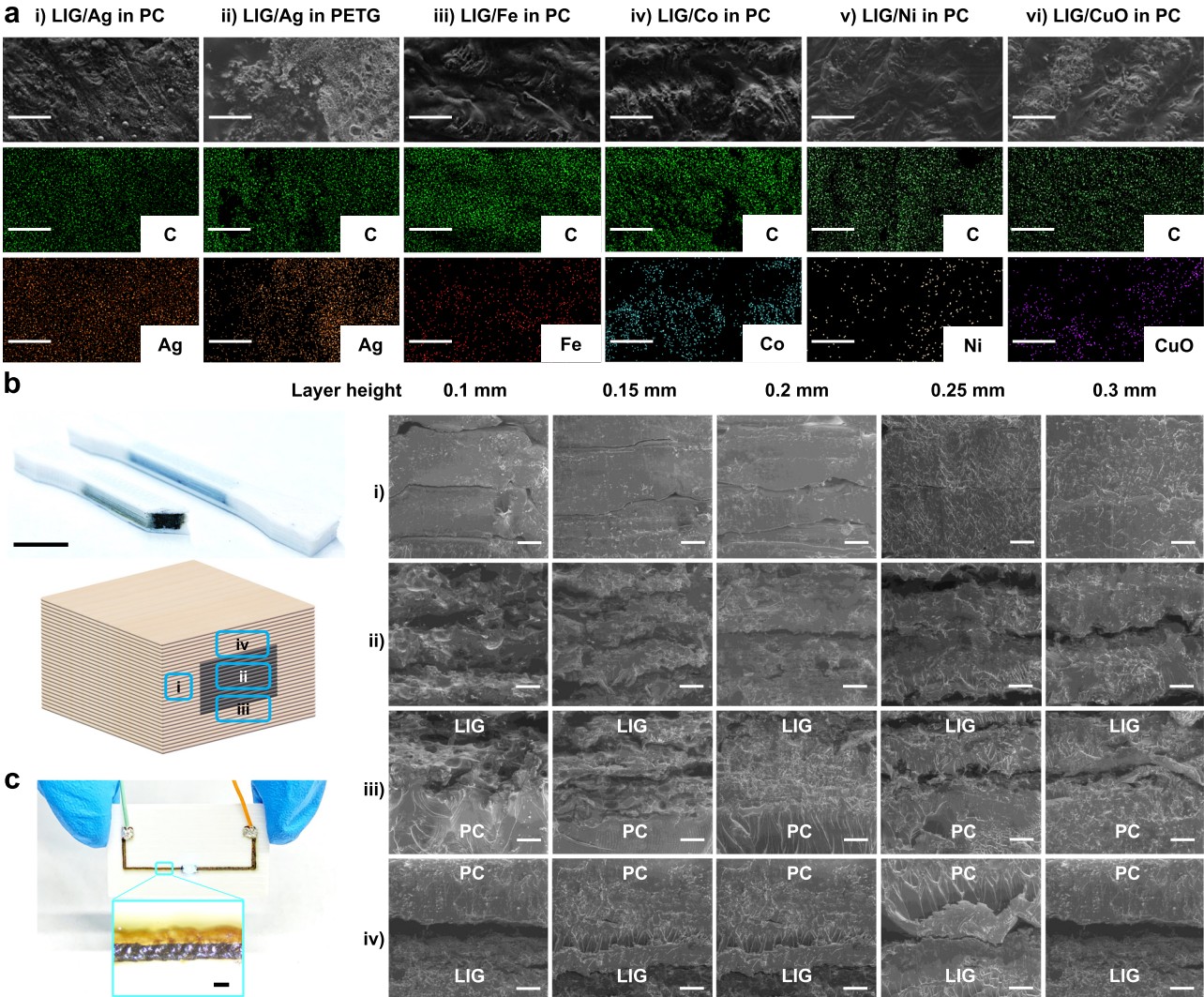

**Fig. 2 | Microscopic characterization. a** SEM and EDS images of metals and metal oxides in LIG induced from various polymers: (i) LIG/Ag in PC; (ii) LIG/Ag in PETG; (iii) LIG/Fe in PC; (iv) LIG/Co in PC; (v) LIG/Ni in PC; (vi) LIG/CuO in PC. Scale bar: 20 μm. **b** Cross-sectional SEM images of LIG produced from PC printed with five different layer heights. They are imaged from four different locations of the 3D structures (denoted with Roman numerals). Scale bar: 10 mm. **c** Photograph of a LIG/Ag electrode to light up a LED. Scale bar: 200 μm.

strength and fracture strain decrease. Figure 3a-ii shows that the sheet resistance of LIG/Ag is superior to that of LIG, reaching as low as 12.36 Ω/sq at a laser power of 2.75 W. Raman spectra were collected from LIG formed from PC using four laser powers (Supplementary Fig. 8). They all display characteristic peaks at ~1330 cm⁻¹, ~1580 cm⁻¹, and ~2700 cm⁻¹, corresponding to the D, G, and 2D bands of a graphitic material, respectively (Fig. 3a-iii). The calculated $I_G/I_D$ ratios is close to 1.5, indicating a low defect level (upper panel of Fig. 3a-iv). Crystallinity sizes (La, in nm), deduced from the $I_G/I_D$ ratios, reach > 60 nm at a laser power of 2.5 W (lower panel of Fig. 3a-iv). To showcase the potential of FMAP, complex 3D structures with spatially patterned LIG were fabricated. These included a gyroid, a Schwarz P surface, a spaceship, and a helix structure (Fig. 3b). The versatility in fabricating complex 3D functional patterns within or on the surfaces of the printed 3D structures was further demonstrated by creation of a 'MU' LIG logo enveloped with a PVDF shell, an airfoil structure embedded with LIG, a 3D lattice embedded within a cuboid, a 3D LIG gear, a 3D LIG fan (Supplementary Fig. 9), and a 3D LIG 'MU' logo with the 'U' part enveloped inside PC and the 'M' part patterned on the outer surface of the structure (Supplementary Fig. 10 and Supplementary Movie 4).

Figure 4 showcases functional materials used as conductive electrodes for PCBs. Examples of a crossbar circuit for LED display and self-capacitance sensors on both rigid and flexible substrates for HMI were demonstrated to show the potential of FMAP in fabricating integrated 3D electronic devices. It shows that compared to traditional PCB fabrication processes that involve chemical etching, our FMAP simplifies the procedures with material utilization of ~100%. A crossbar circuit is a type of a grid-like architecture that uses crossed electrode lines in separate vertical layers. Intersections of these lines create nodes to which devices are connected[10,31]. A crossbar circuit for LEDs offers an advantage by addressing an individual LED, thus lowering power consumption and enhancing the overall energy efficiency of the LED display. An equivalent circuit for a 5×5 LED array and its controller is presented in (Fig. 4a-i), where the anodes and cathodes of the LEDs are connected to the bottom and top electrode lines which are insulated by the printed polymers. To fabricate such a crossbar array, multiple steps by FMAP as shown in Fig. 4a-ii and Supplementary Fig. 11 are deployed. It begins with the FFF printing of a bottom PC layer, which is selectively induced to LIG/Ag electrode lines. Then, the laser selectively induces LIG/Ag electrodes and connection points for the anodes of the LEDs to connect to the bottom electrode. After the encapsulation layer is superimposed over the electrodes by FFF, another laser induction of the top LIG/Ag electrode lines and respective connection points follows for the cathodes of the LEDs to connect

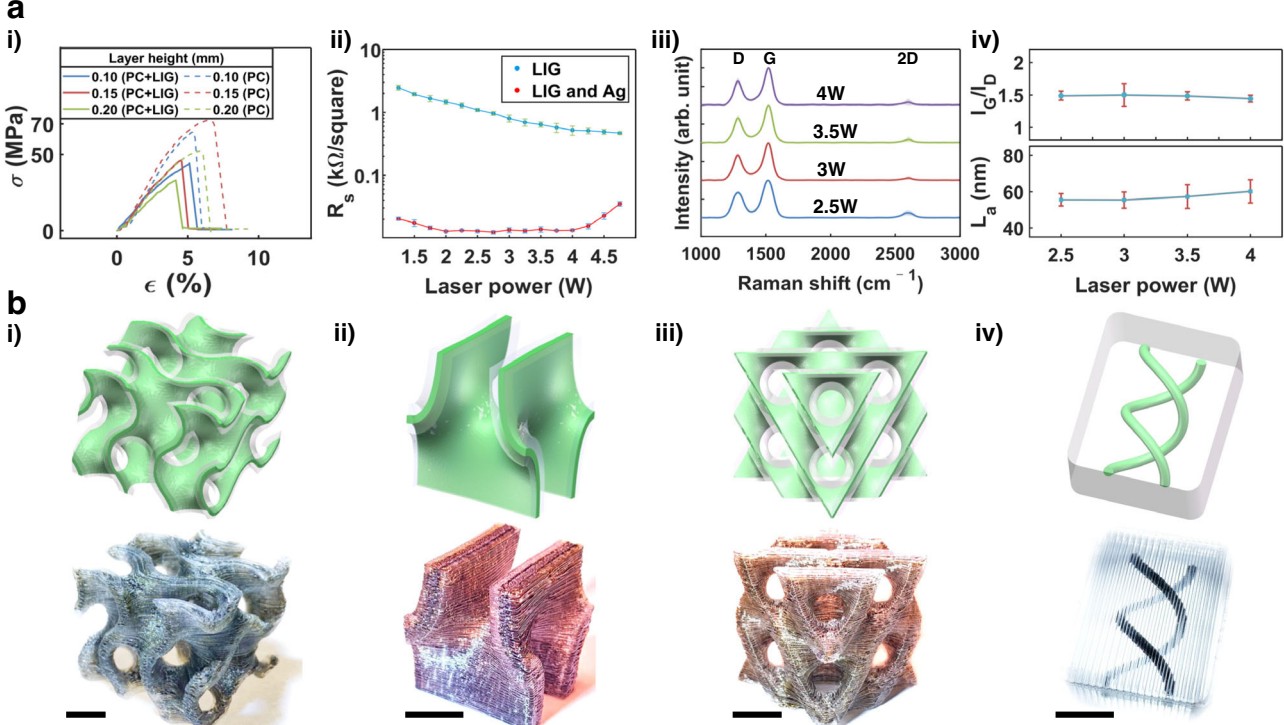

**Fig. 3 | Material property characterization. a** Properties of LIG and LIG/Ag composite in PC: (i) stress-strain curves; (ii) electrical conductivity of LIG and LIG/Ag composite produced at different laser powers. Error bars indicate the standard deviation obtained from 5 sheet resistance measurements; (iii) Raman spectra of LIG produced at different laser powers; (iv) statistical analysis on the ratios of $I_G/I_D$ (upper panel) and calculated average LIG domain sizes (lower panel). Error bars indicate the standard deviation obtained from 10 Raman spectra. **b** Different 3D structures printed from PC with encased LIG inside: (i) a gyroid; (ii) a Schwarz P surface; (iii) A schwarz diamond surface; (iv) a helix. Scale bar: 10 mm.

to the top electrodes. Finally, the LEDs, microcontroller, resistors, capacitors, and crystal are assembled to the nodes to obtain a 5×5 crossbar LED array (Fig. 4a-iii). It demonstrates a capability of controlling an individual LED to display patterns of 'HELLO' (Fig. 4a-iv and Supplementary Movie 5).

Use of touch as an input method for HMI has gained much popularity[32]. It enables users to interact with electronic devices through physical contact with touch-sensitive sensors, e.g., a self-capacitive sensor which is commonly employed due to its ease of implementation and high reliability[33]. It consists of nine conductive electrodes, and the environment serves as a virtual ground. When an object touches the sensing electrode, it modifies the electric field around the electrode, leading to a change in the capacitance[34]. Fabrication of a touchpad with nine capacitive sensing electrodes begins with FFF printing a PETG stencil for all electrodes (Fig. 4b-i). Then a LIG/Ag precursor is deposited by DIW into the stencil followed by the laser induction to form the electrodes. Finally, an encapsulation layer is applied over the electrodes by FFF. Then they are connected to a microcontroller for sensing and wireless communication control. When three fingers touch No. 1, 5 and 9 electrodes, they show > 20% change in their capacitances while others exhibit negligible change (Fig. 4b-ii). This touchpad can be used to control other devices such as a LED array through Bluetooth low energy (BLE) (Fig. 4b-iii and Supplementary Movie 6). Paramount parameters such as the encapsulation thickness and electrode dimensions that affect the capacitance response were investigated. The results are concluded in Supplementary Fig. 12. If using a flexible polymer such as TPU, a flexible touchpad can be fabricated (Fig. 4b-iv).

A slider based on two LIG/Ag triangular electrodes was fabricated using TPU as the structure material (Fig. 4c-i). When the finger slides from the leftmost end to the rightmost end of the slider, the overlapping area between the finger and electrode 1 initially reaches its maximum, then gradually decreases. Consequently, the capacitance of electrode 1 first reaches its maximum and then decreases, while electrode 2 follows an opposite trend with gradual increase to the maximum. By subtracting the normalized data of electrode 1 from electrode 2, the capacitance change of the two electrodes is quite linear to the finger locations (inset of Fig. 4c-ii). Since it is flexible, it can conform to the flat, concave, convex, and curved surfaces (Fig. 4c-iii). With the determined finger position serving as a continuous input signal, it can be used to control the brightness of LEDs (Fig. 4c-iii and Supplementary Movie 7). We then studied the effect of the curvature on the sensor performance. Figure 4c-iv shows that the capacitance change only slightly decreases from 73.4% to 66.6% as the bending curvature increases from 0 to 2.75, highlighting the high flexibility of the device.

Conformable or flexible electronics fabrication are usually fabricated on planar substrates by lithography and then transferred to target substrates[35], resulting in devices confined to the outer surfaces[36]. To demonstrate the capability of FMAP in fabricating functioning devices within 3D structures without lithography or transferring, a ZnO ultraviolet (UV) sensor, a LIG embedded strain-sensing spring, a close-looped haptic robotic manipulator, and a 3D electromagnet, are demonstrated in Fig. 5. A UV sensor measures the environmental UV index[37]. The fabrication begins with FFF printing of a 3D PETG stencil (Supplementary Fig. 13), followed by DIW and FLI to fabricate Ag electrodes in the stencil. ZnO is used as the UV sensing material. Electrical components including a NE555 IC, a capacitor, a resistor, and a LED were integrated into the fabricated 3D circuit (Fig. 5a-i). When there is no UV stimulus, the resistance of ZnO is too large to be measured. When the UV light intensity increases from 130 μW/cm² to 1075 μW/cm², due to the generated charge carriers

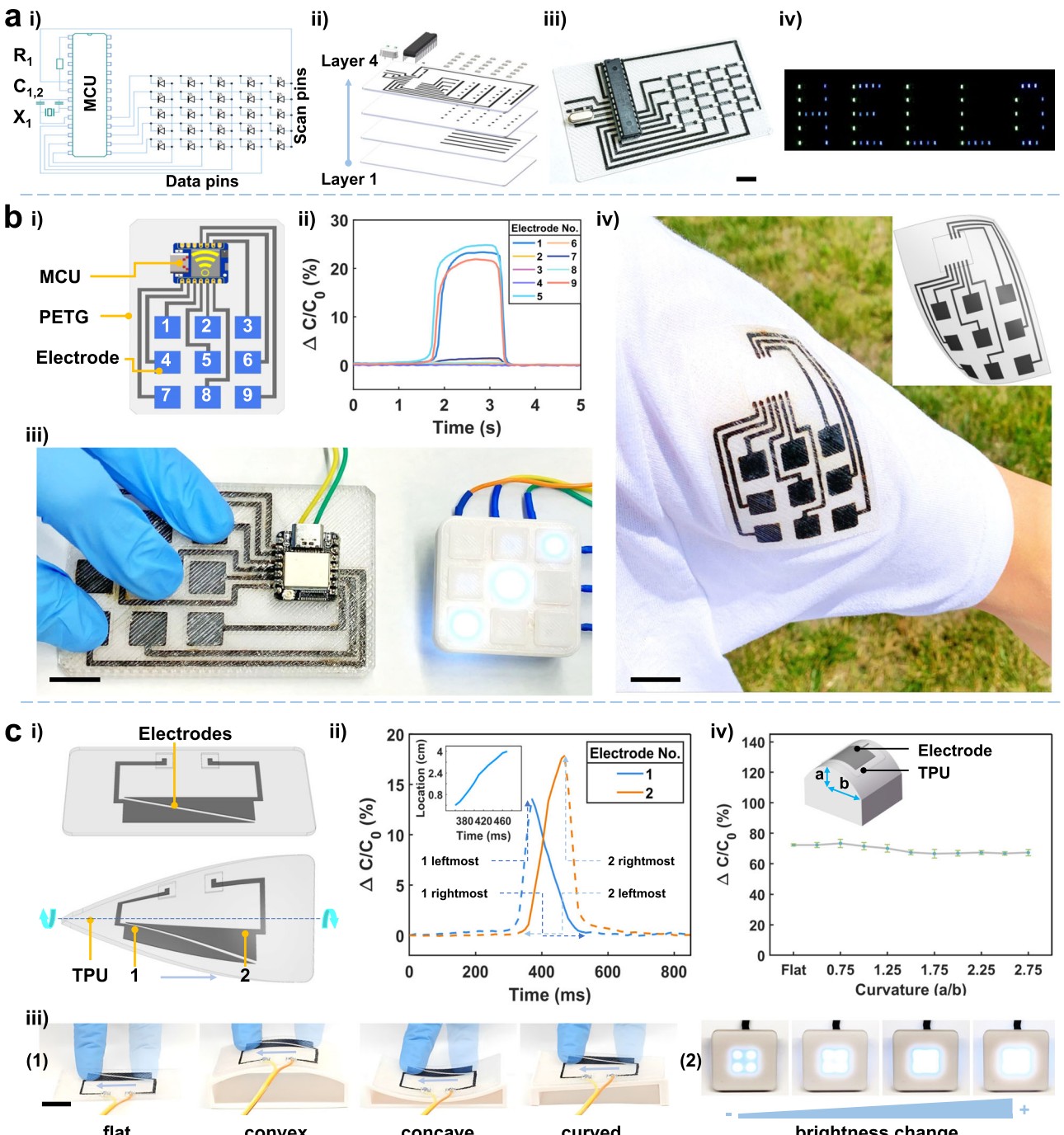

**Fig. 4 | Fabrication of a crossbar circuit for LED array and self-capacitance touch input device by FMAP. a-i** Schematic showing the equivalent circuit of the crossbar LED array and its onboard microchip controller. **a-ii** Exploded view showing the layer-by-layer electrode structure of the crossbar circuit for a LED array. **a-iii** A photograph of the crossbar LED array and its onboard microchip on PC with LIG/Ag as the electrode. **a-iv** Photographs showing the LED array displaying letters of "HELLO". Scale bar: 2 mm. (**b-i**) A schematic showing layout of the touchpad, featuring a PETG substrate, 9 LIG/Ag electrodes, and a microcontroller. **b-ii, iii** Capacitive response and corresponding LED lights when No. 1, 5 and 9

electrodes were touched. **b-iv** Electrodes made from LIG and Ag embedded in TPU printed on textile. Scale bar: 10 mm. **c-i** Layout of a slider featuring two LIG/Ag triangular electrodes packaged inside TPU. As the finger slides from Electrode 1 to Electrode 2, the triangular electrodes facilitate a linear change in contact area on both electrodes, resulting in a linear change in capacitance. **c-ii, iii** Capacitive response of sliders conformed to four types of surfaces as the finger moves between two ends for controlling brightness of a LED. **c-iv** Capacitance change of the slider under different bending curvatures. Error bars indicate the standard deviation obtained from > 10 capacitance measurements. Scale bar: 10 mm.

under the UV irradiation, the total resistance is dramatically reduced (Supplementary Fig. 14). The corresponding photocurrents (I) at an applied bias of 3 V versus the UV light intensities (P) are plotted in Fig. 5a-ii, from which their linear relationship of $\ln(I) = 0.98\ln(P) - 25.3$ is derived with a $R^2$ value of 0.99. This high linearity suggests an

accurate and robust sensing outcome[38]. Then the designed circuit with a LED can indicate the UV intensity change. In this scenario, the NE555 IC acts as an oscillator to produce a square wave output. The fluctuating resistance of the UV sensor leads to a change in the frequency of the generated waveform, thereby affecting the blinking frequency of

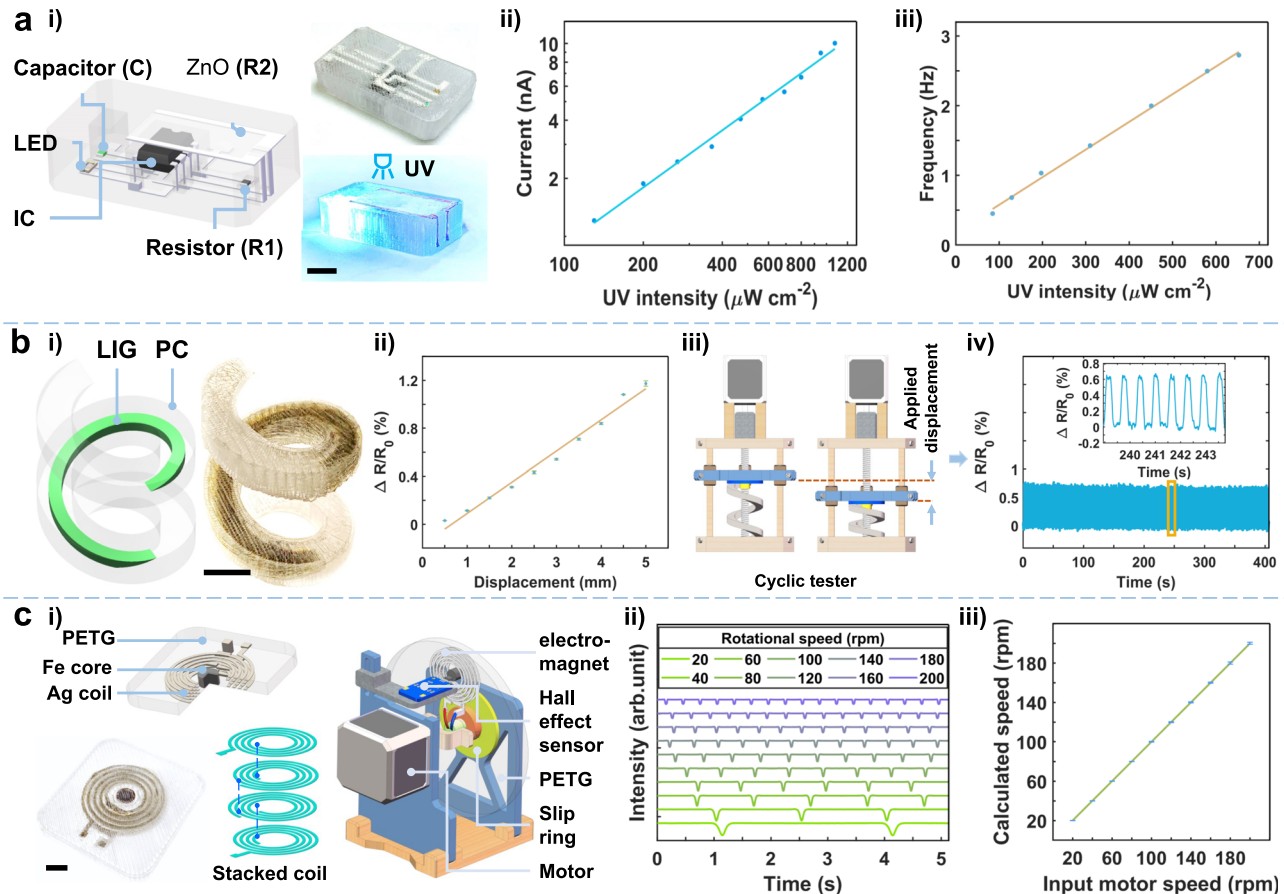

**Fig. 5 | Fabrication of 3D engineered structures with integrated functional devices by FMAP. a**-i Schematic of an integrated UV sensor with electrical components, photographs of the as-fabricated device and the one under UV light. Scale bar: 10 mm. **a**-ii Photocurrents vs. the UV light intensities at a bias of 3 V. **a**-iii On-off frequency as a function of UV intensity. **b**-i A schematic and a photograph of a spring with a PC shell and a LIG core. Scale bar: 10 mm. **b**-ii LIG resistance change as a function of displacement. Error bars indicate the standard deviation obtained from 5 resistance measurements. **b**-iii A scheme showing cyclic testing on the spring. **b**-iv Evolution of resistance change in 640 loading-unloading cycles. **c**-i Schematic and a photograph of a micro electromagnet, its 4-layer coil structure, and its application as an encoder for rotational speed measurement. **c**-ii Hall effect sensor response data at different motor speeds. **c**-iii A plot of the rotation speeds calculated from the Hall effect sensor response data versus the input motor speeds. Error bars indicate the standard deviation obtained from > 20 rotational speed measurements. Scale bar: 10 mm.

the LED. Theoretically, the on and off durations of the LED can be determined by $t_{on} = 0.69 \times C \times R_2$ and $t_{off} = 0.69 \times C \times (R_1 + R_2)$. In this context, C and $R_1$ represent the values for a capacitor and a resistor, and $R_2$ represents the resistance of ZnO. It shows that the on-off switching frequency is quite linear to the UV intensity at a relationship of $f = 3.9 \times 10^{-3} P + 0.18$ with a $R^2$ value of 0.99 (Fig. 5a-iii). The observation of the change of the frequency as the UV intensity increases as shown in Supplementary Movie 8.

A helical compression spring can be used for shock absorption and vibration isolation. To realize a close-loop control, monitoring the loading is necessary, which is traditionally done by attaching a displacement sensor to the spring's surface, but it tends to be inaccurate and quickly worn out[39]. To tackle this issue, we deployed the FMAP to print a functional spring from PC integrated with a LIG strain sensor (Fig. 5b-i). The process involves FFF of PC whose center of each layer (layer height: 0.12 mm) is integrated into LIG. As shown in Fig. 5b-ii and Supplementary Movie 9, the resistance change of the LIG ($\Delta R/R_0$) is quite linear ($R^2 = 0.98$) to the applied displacement (D), from which their relationship is determined to be: $\frac{\Delta R}{R_0} = 2.6 \times 10^{-3} D - 1.7 \times 10^{-3}$. The finite element simulation (FEA) reveals the Von Mises stress distribution at different applied displacements to the spring (Supplementary Fig. 15). Moreover, cycling testing, as shown in Fig. 5b-iii, by applying 3 mm displacement to the spring illustrates that the resistance change signals are consistent after 640 cycles, indicating high

durability of the device (Fig. 5b-iv). To broaden its potential in a robotic application, a gripper made from PC with an embedded LIG strain sensor was fabricated by FMAP (Supplementary Fig. 16a). This integrated gripper can sense force during manipulation, thus enhancing the robot's ability to grab a delicate object in a close-loop manner (Supplementary Fig. 16b-d, Supplementary Movie 10).

An electromagnet is usually made of a coil where an electric current is passed for precise control of the magnetic field, thus is applied in an electrical motor, a magnetic resonance imaging machine, and a robot. When applied to a robot, it can provide accurate and reliable feedback on the position and motion to realize a closed-loop control[40]. We used FMAP to fabricate a micro electromagnet made from a laser-induced 4-layer Ag coil, and a laser-induced iron core in the middle of the coil, both of which are encapsulated by printed PETG (Fig. 5c-i). Note that although the Ag coil is placed vertically on four separate PETG layers, it is electrically connected, and the connection points are illustrated by blue dashed lines. The magnetic field of the electromagnet is primarily determined by the current flowing through the coil, the number of turns in the coil, and the choice of core material. Increasing the current can create excessive Joule heat. Increasing the number of the coil turns increases the space. Thus, using the multi-layered coil as an inductor and the iron as the core material can well-satisfy the design constraints. The micro electromagnet is fabricated together with a rotating disk used for rotational speed measurement

(Fig. 5c-i). When a continuous current passes the Ag coil, a magnetic field is generated, which can be sensed by a stationary Hall effect sensor when the electromagnet approaches. As shown in Supplementary Fig. 17, when the coil is switched on, the voltage of the Hall sensor quickly decreases from 2.4 V to 2.34 V within 0.2 s and returns to 2.4 V within 0.5 s when the power is switched off. When the rotation speeds of the motor which carries the rotational disk increase from 20 revolutions per minute (rpm) to 200 rpm, the time interval between two consecutive voltage spikes—indication of time taken for a turn—is shortened from 2995 ms to 299 ms (Fig. 5c-ii and Supplementary Movie 11). Correspondingly, the rotational speeds can be calculated from these time intervals. They agree well with the input rotational speeds with a $R^2$ of nearly 1.0 (Fig. 5c-iii), showing the high sensing fidelity. This success shows the potential for using FMAP to fabricate highly integrated magnetic devices in robotics with a closed-loop control scheme.

Microfluidic flow reactors have attracted substantial attention due to their ability to better control with high efficiency chemical reactions when compared to traditional bulk vessel reactors[41]. In our previous study, we demonstrated a 3D printed microfluidic reactor for synthesizing zeolitic imidazolate framework (ZIF) NPs with reduced reagent usage, fast reaction rate, and energy savings[42]. However, its operation is limited to room temperature, restricting the range of materials that can be synthesized. An embedded heating electrode for in-situ Joule heating could overcome this limitation[43]. Herein, we demonstrate here the use of FMAP to one-step fabricate an integrated microfluidic reactor with a LIG electrode embedded 0.6 mm underneath the channels as a Joule heater (Fig. 6a, b). Two precursors for ZIF-8 synthesis were fed into the inlets of the channels which were heated by the LIG at a DC voltage of 30 V and a current of 0.1 A. The generated heat accelerates the reaction. To visualize the temperature distribution within the heated channels at various flowing rates (4.5, 9, and 18 μL/s), thermal images were captured using an IR camera (Fig. 6c). When there is no liquid flowing through the channels, the temperature can reach 101 °C. In contrast, as the flow rate increases from 4.5 μL/s to 18 μL/s, the temperature gradually decreased from 66.4 °C to 57.9 °C because heat is carried away by the flowing liquid. The FEA simulation shows the same trend that a higher flow rate leads to the lower temperatures (Fig. 6d and Supplementary Movie 12). As a comparison, the reactions with the flow rates of 4.5 μL/s, 9 μL/s, and 18 μL/s were conducted at room temperature (RT). The samples collected from the reactions at room temperature and elevated temperatures exhibit obvious appearance differences; the former set are transparent while the later set are translucent (Fig. 6e). It may suggest success of the synthesis by the Joule heating. This hypothesis is confirmed by UV-vis absorbance results (Fig. 6f). The samples synthesized at elevated temperatures exhibit much higher absorbance than the samples synthesized at RT, which show no difference with the baseline curves of the precursors.

The morphologies and crystal structures of the as-synthesized ZIF-8 NPs were further examined by Transmission Electron Microscopy (TEM), and X-ray diffraction (XRD). The TEM image in Fig. 6g shows that they ZIF-8 NPs are decagonal; this is one possible projected view of display a particle with a rhombic dodecahedron morphology. The NPs have a diameter of ~ 200 nm. The XRD pattern exhibits characteristic peaks at 2θ of 7.2°, 10.2°, 12.6°, 14.6°, 16.3°, and 17.9°, corresponding to the (011), (002), (112), (022), (013), and (222) planes of ZIF-8 (Fig. 6h)[44]. The sharp and intense peaks indicate a high crystallinity of ZIF-8 synthesized with the in-situ Joule heating. These findings highlight the efficiency and control afforded by the microfluidic reactor and its in-situ heating capability.

## Discussion
FMAP enables the fabrication and assembly of diverse functional and structural materials into a 3D engineered object. The functional materials encompass laser-processable materials like LIG, metals, and metal oxides. As a concept of demonstration, various applications including crossbar LED circuits, capacitive sensor-based touchpads and sliders for HMI, and a UV sensor, are fabricated and tested. Moreover, a LIG strain sensor-embedded spring, gripper for haptic grasping, and micro 3D electromagnets, were realized. Further expanding the application area, a microfluidic reactor featuring Joule heating was demonstrated. The sensors within these prints consistently exhibit attributes of high linearity, accuracy, and rapid response. Overall, FMAP offers advantages for programmed assembly of both functional and structural materials into 3D engineered objects.

Despite the enormous potential in 3D electronic manufacturing, there remain several improvements in the FMAP to be addressed in future. The first one lies in its processing rate. The current setup requires FLI, DIW, and FFF processes to be operated separately. To enhance efficiency, end-effectors of these processes can be equipped on different robotic manipulators to perform simultaneous, collaborative work. Secondly, although the current laser can achieve ~100 μm linewidth meeting the requirement for most printed wearable electronics, higher resolution can be attained by upgrading the laser system. Last but not the least, while the current work focuses on functional materials for electronic applications, future research of applying this FMAP can be extended to different applications such as robotic manipulation, or incorporating other processes such as aerosol printing[45], to this FMAP to further expanding the materials options.

## Methods
### Chemicals
Polyvinylpyrrolidone (PVP, $M_w = 40000$ g/mol, Millipore-Sigma), polyvinyl alcohol (PVA, $M_w = 146,000–186,00$, Sigma Aldrich), silver nitrate ($AgNO_3$, Fisher), sodium citrate ($Na_3C_6H_5O_7$, Sigma Aldrich), zinc oxide nanowires (ZnO NWs, NWZO01A5, ACS Materials), methanol (Fisher), zinc nitrate hexahydrate ($Zn(NO_3)_2 \cdot 6H_2O$, Aldrich), 2-methylimidazole (MeIM, Fisher), polyimide films (DuPont), lignin (Kraft, Domtar), methyl ethyl ketone (MEK, Sigma Aldrich), and silver paste (Ted Palla) were used as received without further purification. Four types of filaments including polycarbonate (PC, Polymaker), thermoplastic polyurethane (TPU, 95 A, Overture), and polyvinylidene fluoride (PVDF, Fluorx), and polyethylene terephthalate glycol (PETG, Overture) were used for FFF. Deionized water (DI $H_2O$) was used to prepare the precursors solutions.

### Preparation of lignin solution and lignin/silver mixture solution
Kraft lignin was first dissolved in MEK (95 wt%) with a mass ratio of 1:3. Subsequently, 4.0 g of the mixture of Kraft lignin and MEK was dissolved in 10 g of a 2 wt% sodium hydroxide (NaOH) solution. PVA solution was prepared by fully dissolving PVA at 9 wt% in DI water at 90 °C for 1 h. Then, the obtained lignin solution was mixed with the PVA solution in a 1:1 volumetric ratio, and stirred until a dark brown homogeneous mixture was produced. To make the mixture of lignin and silver solution, silver ink precursor was added to the lignin/PVA solution at a volumetric ratio of 3:1. Subsequently, the mixture was thoroughly stirred for 1 h.

### Preparation of silver ink solution
First, Solution A was prepared by dissolving 0.30 g of sodium citrate and 0.025 g of PVP in 10 mL of DI $H_2O$. Solution B was prepared by dissolving 0.52 g of $AgNO_3$ in 8 mL of DI $H_2O$. Solution B was introduced dropwise into Solution A under continuous stirring for 1 h to obtain a silver ink solution.

### Materials characterization
X-ray diffraction was obtained on Rigaku SmartLab with Cu Kα radiation ($\lambda = 0.15406$ nm). Raman spectra were collected on a Renishaw Via Raman spectroscopy. The SEM images and element analysis by EDS

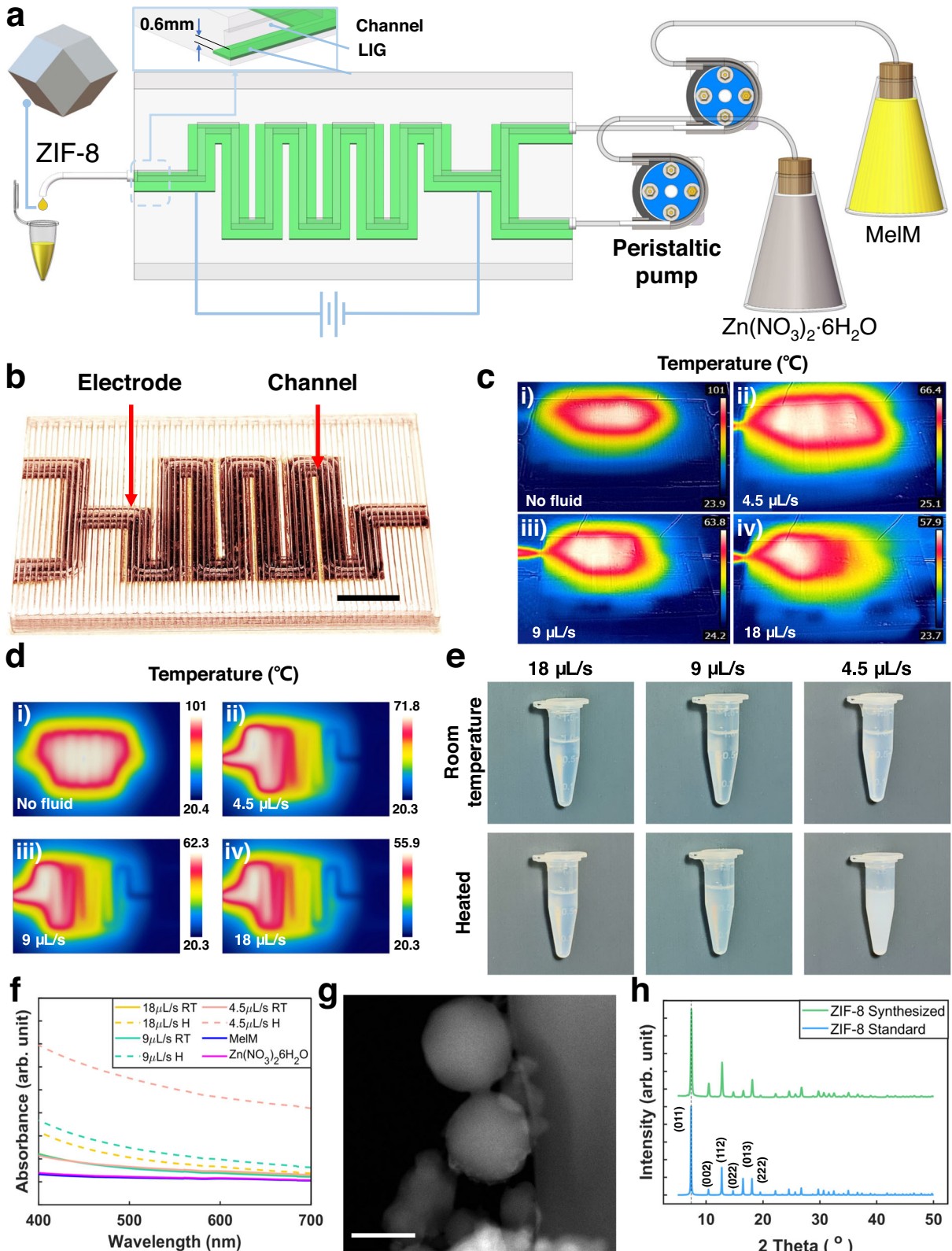

**Fig. 6 | A microfluidic flow reactor with an embedded LIG heater. a** Schematic of an assembled microfluidic flow reactor for ZIF synthesis. **b** Photographs of a fabricated reactor. Scale bar: 10 mm. **c** Thermal images of the reactor taken at different flow rates when power is on. **d** FEA simulation on the temperature distribution of the channels at different flow rates. **e** Photographs of samples synthesized at elevated temperatures and RT. **f** UV–Vis spectra of samples synthesized under RT and Joule heating conditions. **g** A TEM image of synthesized ZIF-8 NPs by in-situ Joule heating at 4.5 μL/s. Scale bar 200 nm. **h** A XRD spectrum of the synthesized ZIF-8 NPs by in-situ Joule heating at 4.5 μL/s and a standard ZIF-8 spectrum.

were taken on the FEI Quanta 400 ESEM FEG system at a voltage of 20 kV.

### Design and construction of the FMAP platform
The FMAP platform was built based on a customized FFF 3D printer (Creality CR-10 V2). The original linear motion mechanism was kept without change. A customized rotational mechanism enabled by two orthogonal NEMA17 stepper motors with harmonic reducers, actuated by DRV8825 at 1/32 micro-stepping, was added. A LASERTREE 5 W laser module, an FFF end, and a DIW syringe were connected to the C-axis actuator. The control system has been modified with Arduino Mega 2560 R3 and a Ramps 1.6+ board. The firmware was modified based on an open-source GRBL-MEGA-5X system.

### Testing of a capacitive-sensing slider
The slider's functionality is calibrated by moving a finger along its surface, spanning from the leftmost end to the rightmost end. To establish a baseline, the minimum and maximum data points were obtained from both electrodes for normalization. Then, the normalized data collected from one electrode was subtracted from the normalized data collected from the other electrode. This subtraction yields linear relationship between the processed capacitance data and the output voltage, hat which in turn controls the brightness of the LED. Wireless communication was made through a pair of Bluetooth devices. Repeated measurements were taken from the same samples for standard deviation calculation.

### Fabrication and testing of a UV sensor
To fabricate a UV sensor, a stencil outlining an Ag circuit was made by FFF of PETG. The silver ink was then deposited onto the stencil by DIW. Subsequently, the silver ink underwent FLI at a scan rate of 500 mm/min and a laser power of 3.5 W to obtain the Ag circuit. After that, electrical components including, a NE555 timer IC, a 10MΩ resistor, a 2.2nf capacitor and a SMD LED were manually assembled into the circuit. Subsequently, 30 μL aqueous solution containing 3 mg/mL ZnO NPs−which serve as the UV sensitive material−was uniformly applied onto the Ag current collector on the top surface the whole packaged device by DIW.

To evaluate the UV sensing capabilities, we employed the AI-2UV20DC UV light source. Adjusting the UV light intensity was accomplished through a shelf that could be raised or lowered, and a UV Meter (General Tools, UV513AB, 280-400 nm) was place together with the UV sensor for UV intensity measurement. Initially, the UV sensor was positioned immediately beneath the UV light source, and the shelf was adjusted to achieve a UV intensity of 1075 μW·cm$^{-2}$. We recorded the sensor's resistance response using a source meter (2604B, Keithley Instruments). Subsequently, we progressively raised the shelf to lower the UV light intensity, eventually reaching 130 μW·cm$^{-2}$. Measurements were taken at each UV intensity level upon stabilization. The performance of the Zno-based UV sensor was assessed based on its alterations in resistance.

### Fabrication and testing of the micro electromagnet
A stencil for later depositing the Ag and Fe was printed from PETG by FFF. The Ag and Fe inks were deposited onto the stencil by DIW, and were then induced to Ag and Fe via FLI a scan rate of 500 mm/min and a laser power of 3.5 W. A 49E linear Hall sensor was used for magnetic field detection. Repeated measurements were taken from the same samples for standard deviation calculation.

### Testing of LIG embedded spring
A customized displacement controller was used to apply various displacements to the spring. The resistance response was measured by a source meter (2604B, Keithley Instruments). Repeated measurements were taken from the same samples for standard deviation calculation.

### Synthesis of ZIF-8 via in-situ heating of microfluidic channel
The current was applied to the LIG electrode by a connected DC power supply (Dr. Meter, HY3005F-3) for Joule heating. The temperature was periodically record by a FLIR E4 camera. After reaching a stable temperature, two precursors of 0.1 M Zn(NO$_3$)$_2$ and 0.8 M MeIM were fed into two inlets of the microfluidic reactor. The obtained products were collected from the outlet. The UV-vis adsorption spectra of the collected samples were measured by a PerkinElmer Lambda 35 UV−vis spectrometer.

### FEA simulation of LIG-embedded spring
The compression test of the spring embedded in LIG was conducted using the commercial software COMSOL Multiphysics. The spring's shell material was defined as Polycarbonate [Solid], sourced from the COMSOL Material Library. A stationary analysis within the "Solid Mechanics" module was executed to determine the Mises stress distribution. This analysis was performed under prescribed displacement at the top surface, with the bottom surface fixed and all other surfaces left free.

### FEA simulation of microfluidic channel heating
The heating of a microfluidic channel with water flow was simulated using the COMSOL Multiphysics software. A stationary analysis, combining "Heat Transfer in Solids and Fluids" and "Laminar Flow", was conducted to compute the temperature distribution. This simulation focused on a heat source positioned within the LIG region, applying a heat rate of 3 W. The microfluidic channel material was specified as Polycarbonate [Solid], while the fluid used within the channel was chosen as Water, a predefined liquid from the COMSOL Material Library. The physical model for water assumed it to be an incompressible Newtonian fluid, and turbulence effects were omitted. The inflow rate ranged from 4.5 μL/s to 18 μL/s, maintaining a constant temperature of 20 °C. At the outlet, a pressure condition of 1 atm and a thermal insulation condition were applied as boundary conditions. Other surfaces were assigned convective heat flux and surface-to-ambient radiation conditions. These conditions were defined with an external temperature of 20 °C, a heat transfer coefficient of 10 W/(m²K), and a surface emissivity of 1. To study the evolution of the temperature field within the channel over time, a time-dependent analysis was carried out with the same boundary conditions as the stationary analysis. The time-dependent simulation began at $t = 0$ s, ended at $t = 300$ s, and used a time step of 1 s.

### Design and construction of FMAP processing apparatus
The platform was built upon a Creality CR-10 V2 3D printer, with the X and Y motion mechanisms upgraded to linear rails driven by a belt. The Z axis is driven by two stepper motors. A customized rotational mechanism, consisting of two orthogonal NEMA17 stepper motors with harmonic reducers from robotdigg, was integrated. Motion control was achieved using an Arduino Mega 2560 and a Ramps 1.6+ board. Temperature control of the FFF hotend and build plate was managed by a separate Creality V2.2 control board.

## Data availability
Source data are provided with this paper.

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

## Acknowledgements

Use of experimental facilities at the MU Electron Microscopy Core (EMC) Facilities is acknowledged. J. L. thanks the financial support from U.S. National Science Foundation (award numbers: 1825352 and 1933861) and U.S. Department of The Interior (grant number: R21AC10073-00). J. L. and J. M. T. acknowledge the financial support by U.S. Army Corps of Engineers, ERDC (grant number: W912HZ-21-2-0050).

## Author contributions

B. Z. designed and constructed the FMAP, and he conducted the structure/device fabrication and characterization. Y. X. contributed to material synthesis and characterization. S. X. and A. C. M. conducted microscopy imaging and analysis. G.H. oversaw the FEA simulation conducted by S.W. and made revisions to the manuscript. Y. W.

conducted tensile testing. S. Y. and C.W. provided the lignin solution and revised the manuscript. J. L. conceived the idea, managed the research progress, and provided regular guidance. J. M. T oversaw the work of S. X. and proofed the manuscript. B. Z. drafted the first manuscript which was thoroughly revised by J. L. All authors commented, revised the manuscript, and agreed on the final version.

## Competing interests

A provisional patent, University of Missouri (MU) application number 24UMC031, owned by MU with inventors of J. L. and B. Z. is to be filed. The remaining authors declare no competing interests.
