## [Peer Review File · Nature Communications]

REVIEWER COMMENTS

Reviewer #1 (Remarks to the Author):

Review of “Programmed Multimaterial Assembly by Synergized 3D Printing and Freeform Laser 2 Induction” by Zheng et al for Nature Comm

ASSESSMENT

This paper focuses on a clever combination of 3D printing and multi-axis lasing; other, similar studies have been carried out in the past, but in this case, there’s a higher degree of control over the angle and placement of the lasing. The authors showcase several examples of how this could work in practical examples.

ISSUES

- Much of the introduction is written in a grandiose fashion as if this process can produce virtually any functional material; this is misleading.
- The main focus of the paper is the FMAP process, but many of the figures are just examples of things that can be produced by FMAP. Are these examples put together systematically? Or is it just random?

MINOR ISSUES

- The poetic living organisms first sentence is out of place in this article.
- “the multimaterial 3D printing” – remove “the”
- Long awkward sentence: “Despite these advancements, challenges persist in these techniques with lacking the versatility in assembling the functional materials within any predesigned locations of the 3D structures and the genericity in broader material choices.”
- Confusing sentence: “For instance, the multi-nozzle DIW and FDM only extrude functional materials that must be blended with polymers with suitable properties” – FDM extrudes polymers.
- Characterizations is not a word. It is “Characterization.”

Reviewer #2 (Remarks to the Author):

In this manuscript, Zheng et al develop a hybrid printing platform for fabricating functional devices. Here, the authors integrated fused filament fabrication (FFF), direct ink writing (DIW), and direct laser writing (DLW) into a five-axis printing platform. In particular, DLW can convert some FFF printed materials into laser-induced graphene and DIW printed ink into functional materials. The authors then demonstrated multiple functional devices that can be directly printed by this new platform. The work is interesting and can be published in Nat. Comm. after the following comments are addressed.

- For the LIG, it was mentioned that layer thickness was about 0.1-0.3mm. Could the authors comment on how well the converted thickness can be controlled?
- In Fig. 1, was LED diode placed manually? In the current text, it sounds that the LED diode was also fabricated.
- For Fig. 2di, did the authors only convert part of the PC samples into LIG? If so, what were the dimension of PC samples (height, width, and length)? Also, what was the dimension of LIG (height, width, and length)? It might be helpful if the authors can also provide the stress-strain curve of printed PC sample.
- FDM is not the official name. Please change to fused filament fabrication (FFF).
- Multimaterial printing for functional applications has drawn significant research interests in recent years. It might be worth to mention some recent advances, such as Nature Communications 14 (1), 1251; Nature Communications 14 (1), 5519. Also, for hybrid printing, photonic curing was used to fabricate structures with conductive traces, for example, Additive Manufacturing 29, 100819.

Response to Reviewers

Reviewer 1

Reviewer #1 (Remarks to the Author):

Review of “Programmed Multimaterial Assembly by Synergized 3D Printing and Freeform Laser 2 Induction” by Zheng et al for Nature Comm

ASSESSMENT

This paper focuses on a clever combination of 3D printing and multi-axis lasing; other, similar studies have been carried out in the past, but in this case, there’s a higher degree of control over the angle and placement of the lasing. The authors showcase several examples of how this could work in practical examples.

Response: we thank the reviewer for the positive assessment of the novelty of our work.

ISSUES

1. Much of the introduction is written in a grandiose fashion as if this process can produce virtually any functional material; this is misleading.

Response: we thank the reviewer for pointing it out, and we have changed the introduction part accordingly, and have removed all information that is misleading. We have updated the introduction part of the manuscript as ‘Assembly of multimaterials into structural and functional components is ubiquitous in nature, inspiring researchers to explore new design principles and fabrication methodologies for creating engineered three-dimensional (3D) structures with multifunctionalities.’^{1,2} Traditionally, hybridized fabrication techniques can be used to achieve the goal,³⁻⁶ but they require multiple, subsequent processes. For instance, producing multilayer 3D printed circuit boards (PCBs) entails steps of etching, lamination, heated pressing, drilling, etc..⁷ The processes require high capital investment while generating unwanted waste streams, thus posing a significant challenge to sustainability. To enhance material utilization efficiency and circumvent the challenge of assembling multimaterials, several new technologies such as mechanics-driven assembly, transfer printing, and multimaterial 3D printing have been emerged recently.^{8,9}

Among these emerging techniques, **the multimaterial 3D printing has attracted attention for its potential benefits, including cost-effectiveness, reduced waste generation, and easy customization.** Thus, many strides have been recently made for applications in multimaterial fabrication.⁹ For example, a direct ink writing (DIW) method enables to fabricate 3D soft electronics⁶ and light

emitting diodes (LEDs).¹⁰ Embedded 3D printing has facilitated production of flexible sensors by embedding functional carbon grease within a polymer encapsulation.¹¹ A multi-nozzle DIW printer with a rapid material switching capability has been developed to print diverse wax-based structures.¹² Further advancement in a core-shell DIW nozzle has enabled assembled multimaterials, such as epoxy/silicone, into different 3D structures, including sandwiches¹³ and helices.¹⁴ Multi-axis fused filament fabrication (FFF)¹⁵ and conformal DIW¹⁶ can make conformal deposition of conductive filaments onto 3D curved surfaces. Moreover, there have been other noteworthy developments in the field, such as digital light processing (DLP) of multimaterials.¹⁷⁻

19

However, within the realm of multimaterial fabrication, these techniques still face challenges of lacking versatility in precisely placing functional materials within 3D structures and access to broader material options. For instance, the embedded 3D printing necessitates preparation of a mold for the structural materials.¹¹ This necessity imposes constraints on the capability of achieving complex geometries, such as in hollow and freestanding features. In the case of core-shell 3D printing, although it can print objects with inner structures made from functional materials, the functional and structural materials are extruded simultaneously and continuously, so depositing the functional materials in predesigned locations, such as outer surface, is not achievable.^{13,14} Besides the limitation in the complexity of printed structures, they often suffer from limited materials options. For instance, the multi-nozzle DIW extrude composite inks that contains both electrically conductive materials and polymers,^{12,15} rendering the resulting materials with low electrical conductivity and low mechanical strength. DLP is quite limited to photosensitive resins.¹⁷ Moreover, the process for multimaterial printing requires switching between different vats while purging non-polymerized residual materials out of the vats, which results in inefficient materials utilization. All these challenges underscore the need for further innovation in the multimaterial fabrication methodologies with improved versatility in the structure complexity and broadened materials choices.

Direct laser writing (DLW) has shown versatility in patterning various functional materials through induced photothermal or/and photochemical effects.²⁰ It greatly expands the library of available materials ranging from laser-induced graphene (LIG),^{21,22} to some metals,²³ metal oxides,²⁴ semiconductors,²⁵ and ceramics.²⁶ A recent trend in DLW is to assemble these functional materials into 3D structures,^{27,28} while this goal is largely limited by its capability in fabricating the functional materials on 2D planes. Recently, we introduced a freeform laser induction (FLI) method facilitated by a 5-axis laser processing platform. This method enables the direct fabrication of 3D conformable electronics on freeform surfaces.²⁹ While this technique represents an advancement in DLW capabilities, spatially patterning functional materials within predesigned locations of 3D structures to create multifunctional objects remains a challenge.

Here, to tackle this challenge, we present a freeform multimaterial assembly process (FMAP) that synergistically marries advantages of three techniques—FLI, DIW, and FFF—to seamlessly assemble both structural and laser-processable functional materials into 3D engineered objects with complex geometries and multifunctionalities. FFF can construct structural components from commercially available thermoplastics such as polycarbonate (PC), polyethylene terephthalate glycol (PETG) and thermoplastic polyurethane (TPU), and polyvinylidene fluoride (PVDF), while FLI selectively converts the FFF-printed material into LIG in predesigned position in the 3D space. DIW can deposit precursors onto LIG electrodes for later laser-inducing other functional materials, e.g., silver, iron, cobalt, nickel, and copper oxides, to obtain LIG-based functional composites. With the advantages of FFF and FLI, the functional materials are either encapsulated inside the printed 3D objects or on their outside surfaces, thus forming integrated functioning 3D devices. They include a crossbar circuit for a light emitting diode (LED) array, strain sensors for an integrated multifunctional spring and a haptic manipulator, a UV sensor, a 3D electromagnet as a rotational encoder, a capacitive sensor for human machine interface (HMI), and an integrated microfluidic reactor with a built-in Joule heater for nanomaterial synthesis. The demonstrated methodology shows a series of advances. Firstly, it facilitates programmed assembly of both functional and structural materials into the integrated 3D devices by a single apparatus, thus eliminating the requirement of many processing steps in different apparatuses. Secondly, it augments the versatility by direct laser processing of different functional materials with negligible precursor waste streams. Thirdly, FLI decouples the synthesis of the functional materials from FFF and DIW, thus it can pattern them in any predesigned locations of the 3D structures. Overall, this methodology represents a step forward in the creation of integrated, multifunctional 3D objects with applications across electronics/sensors, HMI, robotics, and functional microfluids.’

References added:

- 16 Armstrong, C. D. et al. Robotic Conformal Material Extrusion 3D Printing for Appending Structures on Unstructured Surfaces. *Adv. Intell. Syst.*, 2300516 (2024).
- 18 Yue, L. et al. Single-vat single-cure grayscale digital light processing 3D printing of materials with large property difference and high stretchability. *Nat. Commun.* 14, 1251 (2023).
- 19 Yue, L. et al. Cold-programmed shape-morphing structures based on grayscale digital light processing 4D printing. *Nat. Commun.* 14, 5519 (2023).

Also, the first sentence of abstract has been updated as ‘In nature, structural and functional materials often form programmed three-dimensional (3D) assembly to perform daily functions, which has inspired researchers to replicate this principle to engineer these multimaterials into 3D multifunctional structures.’

2. The main focus of the paper is the FMAP process, but many of the figures are just examples of things that can be produced by FMAP.

Response: we appreciate the reviewer's insightful comment. Indeed, the primary focus of our paper is the FMAP process. To illustrate the process, systemic studies on the capabilities of the process must be done. The examples shown in Figure 3-6 are indeed needed to show its broader applicability.

However, we agree with Reviewer that more in-depth study on the material processing would be beneficial to demonstrate the capability of the FMAP process. Thus, besides the results shown in existing Fig. 2, we conducted additional work to obtain new results which are summarized in five supplementary figures to offer a deeper understanding of the capabilities of the FMAP.

The first is to evaluate the laser scribing resolution by characterizing the linewidth of the induced materials from different precursors on diverse polymer substrates. Our observations indicate variations in the linewidth of functional materials based on the precursors and laser parameters (Fig. S6). For instance, the LIG linewidth derived from PETG and PVDF is over 200 μm . If the laser is defocused at -1.5 mm, the Ag linewidth induced from an Ag precursor film on a PC substrate can realize $\sim 100 \mu\text{m}$. These results agree well with our prior findings^{R1} that the influence of the substrate materials and processing variables substantiate the linewidth of laser-induced materials. Nevertheless, for certain applications, e.g., printed wearable electronics, resolution of 100 μm -200 μm is normally sufficient to meet the need.^{R2,R3}

It's worth noting that these results were obtained based on a laser module operated at 450 nm wavelength with an output power of 5 W and a manufacturer-reported focusing spot of 80 μm . To achieve a narrower linewidth, upgrading the laser to a fiber laser with smaller focusing spot is under consideration, as it has been reported to achieve a linewidth of 12 μm for LIG.^{R4}

Figure S6| Resolution of laser induced materials on various substrates. (a) A photograph showing a LIG strip with a 218 μm linewidth induced from lignin on a PETG substrate. **(b)** A photograph showing a LIG strip with a 235 μm linewidth induced from a PVDF substrate, **(c)** A photograph showing a LIG/Ag composite strip with a 247 μm linewidth induced from PC and Ag precursor. **(d)** A photograph showing an Ag strip with a 104 μm linewidth induced from an Ag precursor film on a PC substrate when the laser was defocused at -1.5 mm. Scale bar: 200 μm .

And we added the following sentence in Page 9 of the manuscript: ‘The laser induction resolution is illustrated in Fig. 2c, where a conductive LIG strip with a width of $\sim 200 \mu\text{m}$ can effectively power a LED. **Fig. S6 indicates that the linewidth of the laser induced functional materials varies based on the precursors and laser parameters with the best one achieving $\sim 100 \mu\text{m}$ in the silver.**’

References:

- R1 Zheng, B., Zhao, G., Yan, Z., Xie, Y. & Lin, J. Direct Freeform Laser Fabrication of 3D Conformable Electronics. *Adv. Funct. Mater.* **33**, 2210084 (2023).
- R2 Valentine, A. D. et al. Hybrid 3D Printing of Soft Electronics. *Adv. Mater.* **29**, 1703817 (2017).
- R3 Yu, Y. et al. All-printed soft human-machine interface for robotic physicochemical sensing. *Sci. Robot.* **7**, eabn0495 (2022).
- R4 Stanford, M. G. et al. High-Resolution Laser-Induced Graphene. Flexible Electronics beyond the Visible Limit. *ACS Appl. Mater. Interfaces* **12**, 10902-10907 (2020).

Secondly, we studied how the scan rate of the laser affects the resistivity of the LIG (Fig. S4). It suggests that a slower scan rate results in a smaller resistance, reaching the smallest value of 98.2 Ω/sq at 100 mm/min. Considering the trade-off of increased fabrication time as the decreased scan rate, we chose a scan rate of 300 mm/min, resulting in 163.9 Ω/sq , for LIG fabrication on PC. In addition, we investigated how the LIG thickness varies with the layer height (Fig. S5). It shows that as the laser power rises, the LIG thickness increases. At 1 W, 64 μm thick LIG is converted, reaching 268 μm at 4.5 W.

Figure S4| Sheet resistance of LIG as a function of the laser scan rates. The LIG was induced from PC which was printed by FFF at a 0.15 mm layer height.

Figure S5| Relationship between laser power and the LIG thickness. (a) Photographs showing cross sections of 8 LIG samples made by different laser powers. (b) LIG thickness as a function of laser power. Scale bar: 500 μm .

Accordingly, we have added the following discussion in Page 9 of the manuscript: ‘This observation agrees well with the result shown in Fig. S3, where the electrical resistance in the z-axis direction is dramatically increased when the layer height exceeds 0.15 mm. Fig. S4 shows that a slower scan rate results in a smaller sheet resistance, reaching the smallest value of 98.2 Ω/sq at 100 mm/min. The relationship between the LIG thickness and laser power is revealed in Fig. S5. It shows that as the laser power rises, the LIG thickness increases. The laser induction resolution is illustrated in Fig. 2c, where a conductive LIG trace with a width of 200 μm can effectively power an LED.’

Fig. S7a shows a schematic of LIG distribution within a printed PC sample with thickness of 1 mm and width of 3 mm for tensile testing. Fig. S7b-c shows that as the LIG thickness and width increases, respectively, both the tensile strength and fracture strain decrease. Nevertheless, if we keep the thickness with 0.15 mm or width of 0.1 mm, the tensile strength and fracture stain can be compatible to the pure PC without LIG.

Figure S7| Tensile testing was conducted on PC samples featuring different dimensions of embedded LIG. (a) A schematic illustrates the structure of a tensile testing specimen with LIG embedded inside. (b) Stress-strain curves of PC specimens with varied thicknessed LIG. (c) Stress-strain curves of PC specimens with varied widthed LIG.

Accordingly, we added the following sentences in Page 10 of the manuscript: ‘Furthermore, tensile testing was performed on specimens embedded with LIG. The LIG dimensions were varied while keeping laser power and printing layer height constant. Fig. S7 shows that as the LIG thickness and width increases, respectively, both the tensile strength and fracture strain decrease.’

Finally, to better illustrate the actuation accuracy of the platform, we added new information about the actuation system. In brief, we utilized two stepper motors with harmonic reducers with a reduction ratio of 1:30 to offer higher torque and smoother operation compared to direct-drive systems for the rotational axis, two additional stepper motors for driving Z-axis, and a belt system for driving X and Y axes. These motors were actuated by DRV8825 at 1/32 micro-stepping. To achieve even higher accuracy, we can use DMT542T at 1/128 micro-stepping in future.^{R5}

Correspondingly, we have updated the method part: *Design and Construction of FMAP Processing Apparatus*: The platform was built upon a Creality CR-10 V2 3D printer, with the X and Y motion mechanisms upgraded to linear rails driven by a belt. The Z axis is driven by two stepper motors. A customized rotational mechanism, consisting of two orthogonal NEMA17 stepper motors with harmonic reducers from robotdigg, was integrated. Motion control was achieved using an Arduino Mega 2560 and a Ramps 1.6+ board. Temperature control of the FFF hotend and build plate was managed by a separate Creality V2.2 control board.’

Reference

R5 <https://www.omc-stepperonline.com/download/DM542T.pdf>

Are these examples put together systematically? Or is it just random?

Response: We thank the reviewer for the question. The examples were presented intentionally. To improve the understanding, herein, we explain them in a logical way as follows.

Fig. 3 focuses on functional materials used as conductive electrodes, showcasing applications related to PCBs. We chose this application because, compared to traditional PCB fabrication methods that involve etching, our FMAP’s material utilization rate is close to 100%. This efficiency stems from the minimal waste of substrate material printed by FFF. Moreover, the laser can directly convert the substrate material into conductive electrodes. Therefore, Fig. 3 shows examples of a cross-bar circuit for LED display and self-capacitance sensors on both rigid and flexible substrates for human-machine interfaces.

Accordingly, we have updated the description of Fig. 3 in Page 12: ‘**Fig. 3** showcases functional materials used as conductive electrodes for PCBs. Examples of a cross-bar circuit for LED display and self-capacitance sensors on both rigid and flexible substrates for HMI were demonstrated to show the potential of FMAP in fabricating integrated 3D electronic devices. It shows that compared to traditional PCB fabrication processes that involve chemical etching, our FMAP simplifies the procedures with material utilization of ~100%.’

Also, to further strengthen this logic in presentation, we have upgraded the previous example shown in Fig. 3a by designing a PCB that integrates both a microchip controller and a LED array. Fig. 3a and the corresponding video (Movie S5) for this demonstration were updated. Correspondingly, we updated Fig. S11 for depicting the fabrication steps.

Figure 3| Fabrication of a crossbar circuit for LED array and self-capacitance touch input device by FMAP. (a-i) Schematic showing the equivalent circuit of the crossbar LED array and its onboard microchip controller. **(a-ii)** Exploded view showing the layer-by-layer electrode structure of the crossbar circuit for a LED array. **(a-iii)** A photograph of the crossbar LED array and its onboard microchip on PCB with LIG/Ag as the electrode. **(a-iv)** Photographs showing the

LED array displaying letters of “HELLO”. Scale bar: 2 mm. (b-i) A schematic showing layout of the touchpad, featuring a PETG substrate, 9 LIG/Ag electrodes, and a microcontroller. (b-ii, iii) Capacitive response and corresponding LED lights when No. 1, 5 and 9 electrodes were touched. (b-iv) Electrodes made from LIG and Ag embedded in TPU printed on textile. Scale bar: 10 mm. (c-i) Layout of a slider featuring two LIG/Ag triangular electrodes packaged inside TPU. (c-ii, iii) Capacitive response of sliders conformed to four types of surfaces as the finger moves between two ends for controlling brightness of a LED. (c-iv) Capacitance change of the slider under different bending curvatures. Scale bar: 10 mm.’

Figure R1| Screen shots of updated Movie S5.

Figure S11| Photographs showing fabrication steps for the PCB for the crossbar LED array by FMAP. Scale bar: 10 mm.

Fig. 4 highlights sensors based on functional materials fabricated by FMAP. We chose these

examples to illustrate how FMAP eliminates the need for a transferring process and lithography, allowing for the direct fabrication of functional materials on both exterior and interior surfaces of the target substrates. Previously, devices were typically fabricated on planar substrates and then transferred to the target substrates. By contrast, our FMAP can embed the functional devices inside 3D objects. Herein, we selected ZnO for UV sensing, LIG embedded in a printed spring for strain sensing, and a 3D electromagnet that integrates Fe-core and Ag-coil as part of a magnetic encoder. Additionally, Fig. S16 demonstrates a one-step fabricated robotic gripper with force feedback capability enabled by an LIG strain sensor, providing further insights into the versatility of FMAP.

Accordingly, we have updated the description of Fig. 4 in Page 17: ‘Conformable or flexible electronics fabrication are usually fabricated on planar substrates by lithography and then transferred to target substrates,³⁵ resulting in devices confined to the outer surfaces.³⁶ To demonstrate the capability of FMAP in fabricating functioning devices within 3D structures without lithography or transferring, a ZnO ultraviolet (UV) sensor, a LIG embedded strain-sensing spring, a close-looped haptic robotic manipulator, and a 3D electromagnet, are demonstrated in Fig. 4.’

References added:

- 35 Choi, J. et al. Customizable, conformal, and stretchable 3D electronics via predistorted pattern generation and thermoforming. *Sci. Adv.* **7**, eabj0694 (2021).
- 36 Ershad, F. et al. Ultra-conformal drawn-on-skin electronics for multifunctional motion artifact-free sensing and point-of-care treatment. *Nat. Commun.* **11**, 3823 (2020).

Figure 5 demonstrates an application that shows a 3D functioning microfluid for nanomaterial synthesis. The embedded LIG acts as a heating element, which can be manufactured in a single step with the microfluidic channel. In our previous study, we demonstrated a 3D printed microfluidic reactor for synthesizing zeolitic imidazolate framework (ZIF) NPs, achieving reduced reagent usage, faster reaction rates, and energy savings. However, its operation is limited to room temperature, constraining the range of materials that can be synthesized. Integrating micro-heater in the microfluid channels would open new door for nanomaterial synthesis.

According, we have updated the description of Fig. 5 in Page 21: ‘In our previous study, we demonstrated a 3D printed microfluidic reactor for synthesizing zeolitic imidazolate framework (ZIF) NPs with reduced reagent usage, fast reaction rate, and energy savings.⁴² However, its operation is limited to room temperature, restricting the range of materials that can be synthesized. An embedded heating electrode for in-situ Joule heating could overcome this limitation.⁴³ Herein,

we demonstrate the use of FMAP to **one-step** fabricate an integrated microfluidic reactor with a LIG electrode embedded 0.6 mm underneath the channels as a Joule heater (**Fig. 5a-b**). Two precursors for ZIF-8 synthesis were fed into the...’

Moreover, we have revised the discussion section to highlight the advantages of FMAP in fabricating the demonstrated applications, as well as its limitations along with proposed measures for improvement in future.

‘FMAP enables the fabrication and assembly of diverse functional and structural materials into a 3D engineered object. The functional materials encompass **laser-processable materials like** LIG, metals, and metal oxides. As a concept of demonstration, various **applications** including crossbar LED circuits, capacitive sensor-based touchpads and sliders for HMI, and a UV sensor, are fabricated and tested. Moreover, a LIG strain sensor-embedded spring, gripper for haptic grasping, and micro **3D** electromagnets, were realized. Further expanding the application area, a microfluidic reactor featuring Joule heating was demonstrated. The sensors within these prints consistently exhibit attributes of high linearity, accuracy, and rapid response. **Overall, FMAP offers advantages for programmed assembly of both functional and structural materials into 3D engineered objects. Despite the enormous potential in 3D electronic manufacturing, there remain several improvements in the FMAP to be addressed in future. The first one lies in its processing rate. The current setup requires FLI, DIW, and FFF processes to be operated separately. To enhance efficiency, end-effectors of these processes can be equipped to different robotic manipulators to perform simultaneous, collaborative work to improve the processing rate. Secondly, although the current laser can achieve ~100 μm linewidth meeting the requirement for most printed wearable electronics, higher resolution can be attained by upgrading the laser system. Last but not the least, while the current work focuses on functional materials for electronic applications, future research includes extending this FMAP to fields such as robotic fabrication, and incorporating other processes, such as aerosol printing,⁴⁵ to this FMAP to further expanding the materials options.’**

Reference added:

45 Roach, D. J. et al. The m4 3D printer: A multi-material multi-method additive manufacturing platform for future 3D printed structures. *Addit. Manuf.* **29**, 100819 (2019).

MINOR ISSUES

- *The poetic living organisms first sentence is out of place in this article.*

We thank the reviewer for this suggestion, the first sentence of the introduction has been revised to ‘Assembly of multimaterials into structural and functional components is ubiquitous in nature, inspiring researchers to explore new design principles and fabrication methodologies for creating engineered three-dimensional (3D) structures with multifunctionalities.^{1,2}

The first sentence of the abstract part has been revised to ‘In nature, structural and functional materials often form programmed three-dimensional (3D) assembly to perform daily functions, which has inspired researchers to replicate this principle to engineer these multimaterials into 3D multifunctional structures.’

- *“the multimaterial 3D printing” – remove “the”*

We thank the reviewer for such detailed comments. “the” was removed from the sentence.

- *Long awkward sentence: “Despite these advancements, challenges persist in these techniques with lacking the versatility in 61 assembling the functional materials within any predesigned locations of the 3D structures and the 62 genericity in broader material choices.”*

We thank the reviewer for pointing this out, the sentence has been revised to ‘these techniques still face challenges of lacking versatility in precisely placing functional materials within 3D structures and access to broader material options.’

- *Confusing sentence: “For instance, the multi-nozzle DIW and FDM only extrude functional materials that must be blended with polymers with suitable properties” – FDM extrudes polymers.*

We Thank the reviewer for pointing this out, the ‘FDM’ part has been removed and the sentence has been updated as ‘For instance, the multi-nozzle DIW extrude composite inks that contains both electrically conductive materials and polymers,^{12,15} rendering the resulting materials with low electrical conductivity and low mechanical strength.’

- *Characterizations is not a word. It is “Characterization.”*

We thank the reviewer for pointing this out, the ‘Characterizations’ in both the caption of Fig. 2 and Author contributions are corrected.

‘Figure 2| Characterization. (a) SEM and EDS images of metals and metal oxides in LIG induced from various polymers: (i) LIG/Ag in PC; (ii) LIG/Ag in PETG; (iii) LIG/Fe in PC; (iv) LIG/Co in PC; (v) LIG/Ni in PC; (vi) LIG/CuO in PC. Scale bar: 20 μm....’

And ‘...B. Z. designed and constructed the FMAP, and he conducted the structure/device fabrication and **characterization**. Y. X. contributed to material...’

Reviewer 2

In this manuscript, Zheng et al develop a hybrid printing platform for fabricating functional devices. Here, the authors integrated fused filament fabrication (FFF), direct ink writing (DIW), and direct laser writing (DLW) into a five-axis printing platform. In particular, DLW can convert some FFF printed materials into laser-induced graphene and DIW printed ink into functional materials. The authors then demonstrated multiple functional devices that can be directly printed by this new platform. The work is interesting and can be published in Nat. Comm. after the following comments are addressed.

Response: we thank the reviewer for the positive assessment of our work.

For the LIG, it was mentioned that layer thickness was about 0.1-0.3 mm. Could the authors comment on how well the converted thickness can be controlled?

Response: we thank the reviewer for this question, additional experiment has been done to address this issue.

We first studied how the scan rate of the laser affects the sheet resistance of the LIG (Fig. S4). It suggests that a slower scan rate results in a smaller resistance, reaching the smallest value of 98.2 Ω/sq at 100 mm/min. Considering the trade-off with the increased fabrication time as the decreased scan rate, we chose a scan rate of 300 mm/min, resulting in 163.9 Ω/sq , for LIG patterning on PC.

Figure S4| Sheet resistance of LIG as a function of the laser scan rate. The LIG was induced from PC which was printed by FFF at a 0.15 mm layer height.

To characterize the LIG conversion thickness, we varied the laser power from 1 W to 4.5 W while maintaining a scan rate of 300 mm/min to analyze the relationship between laser power

and conversion thickness. The results are summarized in Fig. S5. Fig. S5a displays photographs of cross sections of eight LIG samples patterned by lasers with gradually increased laser power. Fig. S5b illustrates the measured LIG layer thickness. It shows that as the laser power rises, the LIG thickness increases. At 1 W, 64 μm LIG is obtained, reaching 268 μm at 4.5 W. This finding is consistent with the data presented in Fig. S3, which indicates that a laser operated at 2.5 W cannot convert LIG beyond 200 μm .

Figure S5| Relationship between laser power and the LIG thickness. (a) Photographs showing cross sections of 8 LIG samples made by different laser powers. **(b)** LIG thickness as a function of laser power. Scale bar: 500 μm .

And we have included the laser parameters into the caption of Fig. S3 as follows.

Figure S3| Characterization on the LIG induced from PC printed with different layer heights. (a) Cross-sectional optical images of the LIG induced from PC printed with five different layer heights, a scan rate of 300mm/min and power of 2.5 W. Scale bar: 200 μm . **(b)** Change of the resistance of LIG in the z-axis direction vs. the layer height. Note: because of decreased

conductivity along the z-axis as the increased layer height, a layer height of 0.15 is typically employed for all the showcased devices in this study.

Accordingly, we added the following discussion in Page 9 of the manuscript ‘This observation agrees well with the result shown in **Fig. S3**, where the **sheet** resistance in the z-axis direction is dramatically increased when the layer height exceeds 0.15 mm. **Fig. S4 shows that a slower scan rate results in a smaller sheet resistance, reaching the smallest value of 98.2 Ω /sq at 100 mm/min. The relationship between the LIG thickness and laser power is revealed in Fig. S5. It shows that as the laser power rises, the LIG thickness increases.** The laser induction resolution is illustrated in **Fig. 2c**, where a conductive LIG **electrode** with a width of 200 μm can effectively power an LED.’

In Fig. 1, was LED diode placed manually? In the current text, it sounds that the LED diode was also fabricated.

Response: we thank the reviewer for this question. Yes, the LED diode was assembled manually. To avoid confusion, we have updated the related sentences in Page 10 of the manuscript. ‘Fig. 1d displays the fabricated wireless LED, with a cross-section view illustrating the distribution of the conductive LIG/Ag electrode inside the PC structure. Note that the LED diode was manually integrated with the FMAP-fabricated coil.’

For Fig. 2d-i, did the authors only convert part of the PC samples into LIG? If so, what were the dimension of PC samples (height, width, and length)? Also, what was the dimension of LIG (height, width, and length)? It might be helpful if the authors can also provide the stress-strain curve of printed PC sample.

Response: we thank the reviewer for this question. Yes, only part of the PC was converted to LIG. The corresponding description in Page 10 has been updated as ‘Tensile testing specimens (dimensions: 25 mm \times 3 mm \times 1 mm) with embedded LIG in the center (dimensions: 25 mm \times 2mm \times 0.4 mm) were produced by our FMAP. The PC was printed with the layer heights of 0.1-0.2 mm. Fig. 2d-i shows that their tensile strengths all exceed 35 MPa, which is compatible to pure PC specimens, indicating well-maintained mechanical properties even if the PC is partially converted to LIG.’

We have included the stress-strain curve of the printed PC samples into Fig. 2d-i.

Accordingly, Fig.2 has been revised as follows.

Figure 2| Characterization. (a) SEM and EDS images of metals and metal oxides in LIG induced from various polymers: (i) LIG/Ag in PC; (ii) LIG/Ag in PETG; (iii) LIG/Fe in PC; (iv) LIG/Co in PC; (v) LIG/Ni in PC; (vi) LIG/CuO in PC. Scale bar: 20 μ m. (b) Cross-sectional SEM images

of LIG produced from PC printed with five different layer heights. They are imaged from four different locations of the 3D structures. Scale bar: 10 mm. (c) Photograph of a LIG/Ag electrode to light up a LED. Scale bar: 200 μm . (d) Properties of LIG and LIG/Ag composite in PC: (i) stress-strain curves; (ii) electrical conductivity of LIG and LIG/Ag composite produced at different laser powers; (iii) Raman spectra of LIG produced at different laser powers; (iv) statistical analysis on the ratios of I_G/I_D (upper panel) and calculated average LIG domain sizes (lower panel). (e) Different 3D structures printed from PC with encased LIG inside: (i) a gyroid; (ii) a Schwarz P surface; (iii) a spaceship; (iv) a helix. Scale bar: 10 mm.

Also, we have done additional experiment to characterize the thickness and width of LIG, and the results are concluded in Fig. S7.

Figure S7| Tensile testing was conducted on PC samples featuring different dimensions of embedded LIG. (a) A schematic illustrates the structure of a tensile testing specimen with LIG embedded inside. **(b)** Stress-strain curves of PC specimens with varied thickened LIG. **(c)** Stress-strain curves of PC specimens with varied widened LIG.

Accordingly, we added the following sentences in Page 10 of the manuscript: ‘Furthermore, tensile testing was performed on specimens embedded with LIG. The LIG dimensions were varied while keeping laser power and printing layer height constant. Fig. S7 shows that as the LIG thickness and width increases, respectively, both the tensile strength and fracture strain decrease.’

FDM is not the official name. Please change to fused filament fabrication (FFF).

Response: we thank the reviewer for the suggestion. All 33 ‘FDM’ and ‘fused deposition

modeling' in both manuscript and supporting information have been corrected to 'FFF' and 'fused filament fabrication', respectively. And the 'FDM' in Fig. 1c and Fig. S1 has also been corrected.

Figure 1| Schematic of FMAP and workflow of fabricating 3D devices by assembling structural and functional materials using FMAP. (a) Schematic showing FMAP and its actuation system. **(b)** Schematic of end effectors for FFF, DIW, and FLI. **(c)** Workflow of fabricating a 3D wireless LED circuit with LIG (induced from PC) and Ag electrodes. **(d)** Scheme and a photograph of the fabricated 3D wireless LED. **(e)** A photograph of a fabricated 3D wireless LED with LIG (induced from lignin) and Ag electrodes on a cloth. Scale bar: 10 mm.

Figure S1| Workflow of fabricating a wireless LED using FMAP. (a) Design and modeling of electrodes for a wireless LED and respective G-code generation. The coil and shell electrodes are modeled independently. Subsequently, the toolpaths for both the coil and shell are generated using a slicer tool, which are tailored to the specific requirements of FLI, DIW, and FFF. (b) Toolpaths for FFF, FLI, and DIW. To achieve precise positioning of the three end effectors, the toolpaths are integrated with their respective offset parameters. (c) Time-lapse images of the different fabrication steps for a wireless LED: (i) printing of a polycarbonate (PC) substrate; (ii) multilayer LIG/Ag electrodes are fabricated within the printed PC structure; (iii) conformal patterning of the LIG/Ag electrodes on the surface of the PC structure. Throughout this procedure, all actuators work synergistically to ensure that the laser beam remains perpendicular to the target 3D surface.’

Multimaterial printing for functional applications has drawn significant research interests in recent years. It might be worth to mention some recent advances, such as Nature Communications 14 (1), 1251; Nature Communications 14 (1), 5519. Also, for hybrid printing, photonic curing was used to fabricate structures with conductive traces, for example, Additive Manufacturing 29, 100819.

Response: We appreciate the reviewers for the information of recent advances in

multimaterial printing. Correspondingly, we include these references in Introduction ‘Multi-axis fused filament fabrication (FFF)¹⁵ and conformal DIW¹⁶ can make conformal deposition of conductive filaments onto 3D curved surfaces.’, ‘such as a modified digital light processing (DLP) printer capable of high-resolution multimaterial printing.¹⁷⁻¹⁹’, **and in Discussion** ‘Last but not the least, while the current work focuses on functional materials for electronic applications, future research includes extending this FMAP to fields such as integrated robot fabrication, and incorporating other processes such as aerosol printing⁴⁵ to this FMAP to further expanding the materials options.’

And we have added the following references:

- 16 Armstrong, C. D. et al. Robotic Conformal Material Extrusion 3D Printing for Appending Structures on Unstructured Surfaces. *Adv. Intell. Syst.*, 2300516 (2024).
- 18 Yue, L. et al. Single-vat single-cure grayscale digital light processing 3D printing of materials with large property difference and high stretchability. *Nat. Commun.* 14, 1251 (2023).
- 19 Yue, L. et al. Cold-programmed shape-morphing structures based on grayscale digital light processing 4D printing. *Nat. Commun.* 14, 5519 (2023).
- 45 Roach, D. J. et al. The m4 3D printer: A multi-material multi-method additive manufacturing platform for future 3D printed structures. *Addit. Manuf.* 29, 100819 (2019).

REVIEWERS' COMMENTS

Reviewer #1 (Remarks to the Author):

The authors have addressed my concerns. I appreciate how thorough their response was.

Reviewer #2 (Remarks to the Author):

The authors have addressed my comments in my previous review and the paper can be accepted.

Response to Reviewers

Reviewer 1

The authors have addressed my concerns. I appreciate how thorough their response was.

Response: we thank the reviewer for the positive assessment of our revised work.

Reviewer 2

The authors have addressed my comments in my previous review and the paper can be accepted.

Response: we thank the reviewer for the positive assessment of our revised work.